# What, how and who: Cost-effectiveness analyses of COVID-19 vaccination to inform key policies in Nigeria

**Francis J. Ruiz**[1☯]*, **Sergio Torres-Rueda**[1☯], **Carl A. B. Pearson**[2,3,4], **Eleanor Bergren**[1], **Chinyere Okeke**[5], **Simon R. Procter**[3], **Andres Madriz-Montero**[1], **Mark Jit**[2,3], **Anna Vassall**[1], **Benjamin S. C. Uzochukwu**[5]

1 Department of Global Health & Development, Faculty of Public Health and Policy, London School of Hygiene and Tropical Medicine, London, United Kingdom, 2 Department of Infectious Disease Epidemiology, Faculty of Epidemiology and Population Health, London School of Hygiene and Tropical Medicine, London, United Kingdom, 3 Centre for Mathematical Modelling of Infectious Diseases, London School of Hygiene and Tropical Medicine, London, United Kingdom, 4 South African DSI-NRF Centre of Excellence in Epidemiological Modelling and Analysis, Stellenbosch University, Stellenbosch, Republic of South Africa, 5 Department of Community Medicine, University of Nigeria Nsukka, Enugu Campus, Nsukka, Nigeria

☯ These authors contributed equally to this work.
* Francis.Ruiz@lshtm.ac.uk

**Data Availability Statement:** The CovidM modelling framework has been published previously and is available on the CMMID COVID-

## Abstract

While safe and efficacious COVID-19 vaccines have achieved high coverage in high-income settings, roll-out remains slow in sub-Saharan Africa. By April 2022, Nigeria, a country of over 200 million people, had only distributed 34 million doses. To ensure the optimal use of health resources, cost-effectiveness analyses can inform key policy questions in the health technology assessment process. We carried out several cost-effectiveness analyses exploring different COVID-19 vaccination scenarios in Nigeria. In consultation with Nigerian stakeholders, we addressed three key questions: what vaccines to buy, how to deliver them and what age groups to target. We combined an epidemiological model of virus transmission parameterised with Nigeria specific data with a costing model that incorporated local resource use assumptions and prices, both for vaccine delivery as well as costs associated with care and treatment of COVID-19. Scenarios of vaccination were compared with no vaccination. Incremental cost-effectiveness ratios were estimated in terms of costs per disability-adjusted life years averted and compared to commonly used cost-effectiveness ratios. Viral vector vaccines are cost-effective (or cost saving), particularly when targeting older adults. Despite higher efficacy, vaccines employing mRNA technologies are less cost-effective due to high current dose prices. The method of delivery of vaccines makes little difference to the cost-effectiveness of the vaccine. COVID-19 vaccines can be highly effective and cost-effective (as well as cost-saving), although an important determinant of the latter is the price per dose and the age groups prioritised for vaccination. From a health system perspective, viral vector vaccines may represent most cost-effective choices for Nigeria, although this may change with price negotiation.

19 GitHub page. All code used are publicly available at: https://zenodo.org/badge/latestdoi/323964488.

**Funding:** The contributions of ST-R, FR, CABP, AMM, BSC (Uzochukwu), CO, EB, SRP, MJ and AV are supported by the International Decision Support Initiative, which is funded by the Bill and Melinda Gates Foundation (OPP1202541). MJ has received funding from the European Union's Horizon 2020 research and innovation programme - project EpiPose (Grant agreement number 101003688); MJ has also received funding from the National Institute for Health Research Health Protection Research Unit (NIHR HPRU) in Modelling and Health Economics at Imperial College and LSHTM in partnership with UKHSA. The European Commission is not responsible for any use that may be made of the information it contains. The views expressed are those of the author(s) and not necessarily those of the NHS, the NIHR, the Department of Health or UKHSA. The contributions of CABP is supported by the World Health Organization. The funders had no role in study design, data collection and analysis, decision to publish, or preparation of the manuscript. Funders supported researcher time and other resources (such as computer equipment) needed for completion of the study.

**Competing interests:** The authors have declared that no competing interests exist.

## Introduction

A number of safe and highly efficacious COVID-19 vaccines have been developed and are being rolled out globally; some high-income countries have achieved high-levels of coverage, although in many low- and middle-income country (LMIC) settings access to vaccines have been limited and deployment slow. An important component in informing spending decisions as part of Health Technology Assessment (HTA), even in pandemic settings, is the assessment of cost-effectiveness [1]. Cost-effectiveness analysis has been applied to COVID-19 vaccinations with the aim to inform price negotiations and assess the opportunity cost of allocating scarce funds to COVID-19 prevention in resource poor settings [2]. Even where vaccines are provided to LMICs freely through the COVID-19 Vaccines Global Access (COVAX) initiative or bilateral donations, other resources are used in delivery and opportunity costs remain both globally and locally.

The first case of COVID-19 in Nigeria, a country of over 200 million people, was recorded at the end of February 2020 [3]. Two years later, the official number of reported cases exceeded 250,000 [4]. While reported COVID-19 related deaths total approximately 3000, the real number of deaths could be many times higher; a recent study suggests excess mortality in Nigeria for 2020–21 could surpass 160,000 deaths [5]. Nigeria had only administered about 56 million doses by end of July 2022, with approximately 25 million people having received the full initial vaccination protocol [6]. This represents about 12% of the population being full vaccinated, which is in contrast to the 62% global average [7].

In the context of low uptake, as well as substantial resource constraints in health, Nigeria was one of several countries that informed key national policy priorities on COVID-19 vaccination through evidence informed approaches [8]. Nigerian policy makers and decision authorities, including the Federal Ministry of Health, sought to define key policy questions. An initial long list was narrowed down to three broad areas that could be feasibly explored through transmission dynamic modelling and existing data. The first policy question related to which vaccine type should be purchased, and in particular, to comparing the vaccines from COVAX with other vaccines that may be bought from the market due to limited COVAX supply. Available vaccines vary in terms of efficacy, price per dose, as well as associated delivery costs, such as additional cold chain or storage requirements. The second area concerned the method of delivery. Most existing immunisation programmes in Nigeria cover infants and children. It is therefore not evident what is the financially optimal method of delivery for vaccination that largely targets adults. Further, in a country where about 50% of the population lives in non-urban areas, it is important to understand the cost-effectiveness of non-facility-based methods of delivery [9]. Lastly, given the differential health impacts of COVID-19 by age, as well as the slow uptake of the vaccine, policy makers were interested in knowing how to prioritise delivery across age groups.

To address these three key policy questions, we carried out a set of cost-effectiveness analyses to support decision makers in Nigeria.

## Materials and methods

### Ethics statement

This study–part of a wider health technology assessment exercise–was approved by the University of Nigeria Teaching Hospital Research Ethics Committee (Ref: NHREC/05/01/2008B-FWA00002458-1RB00002323).

Our cost estimation and epidemiological modelling were based on publicly available data sources as well as validation from the authors. As a result, our work did not involve research

participants and therefore did not necessitate obtaining informed consent nor developing specific protocol to ensure data anonymisation.

This paper is the authors' own original work and reflects the authors' own research in a truthful manner. All authors have been personally and actively involved in substantial work leading to the paper and take responsibility for its content. The paper properly credits the contribution of all co-authors and researchers involved. We have cited all sources used accurately.

This research is guided by a desire to inform policy research questions have been constructed accordingly and we have attempted to be as transparent as possible in how we present assumptions. Key data are publicly available. Should readers have any further questions or would like further disaggregation of data, we encourage them to contact the corresponding author.

## Study design

The objective of this study is to evaluate the cost-effectiveness of COVID-19 vaccination in Nigeria taking into account different vaccine alternatives, delivery platforms, and age groups.

We combined an epidemiological model of SARS-CoV-2 transmission (CovidM) [2] parameterised with Nigeria specific data with a costing model that incorporated local resource use assumptions and prices. We compared scenarios of vaccination (intervention) with no vaccination (comparator) over a five-year time horizon from a health sector perspective. We estimated incremental cost-effectiveness ratios (ICERs) in terms of cost per disability-adjusted life year (DALYs) averted and compared these to commonly used cost-effectiveness ratios. The analysis followed the Consolidated Health Economic Evaluation Reporting Standards (CHEERS; see supplementary materials in [2]), and adhered to the iDSI Reference Case [10].

## Scenarios by policy question

The characteristics of each scenario for the three policy questions were defined as follows:

1. **What vaccine to purchase?** In order to reflect the real-world effectiveness of vaccines and recognise the regularly changing evidence base behind individual vaccines, the modelling sought to analyse vaccine types that link with the target vaccines of interest to Nigerian policy makers. Vaccine effectiveness was estimated for two categories of vaccines: (1) viral vector vaccines (AZ- or J&J-like) and (2) mRNA vaccines (Moderna- or Pfizer-BioNTech-like). The vaccine estimates used in the cost-effectiveness analysis should therefore not be interpreted as specifically linked to an individual vaccine product, but rather to be broadly reflective of a vaccine with similar characteristics.

2. **How should vaccines be delivered?** It was assumed that vaccine doses would be delivered through three modes: (1) health facility-based, (2) campaigns (where vaccinators set up a vaccination site outside of a health facility over a period of time) and (3) targeted campaigns (where vaccinators attend locations where people are congregated, such as markets and places of worship, one day at a time).

3. **Who should be prioritized?** Five age-related and target coverage scenarios were explored based on discussions with policy makers: (1) 70% of adults 50 years and above, (2) 100% of adults 50 years and above, (3) 25% of adults (but first prioritising all those aged 50 years and above), (4) 70% of adults 50 years and above, and 25% of those aged 18–49, (5) 90% of all adults aged 18 and above. We modelled costs of delivering one mRNA vaccine (Moderna-like) and one viral vector vaccine (AstraZeneca-like).

## Epidemiological model

To capture the natural history and transmission of SARS-CoV-2, CovidM, a previously published compartmental model, was used [11–13], tailored to the population of Nigeria using data from WorldPop (2019) [14]. See Pearson et al. for further details [2].

The model compartments are an extended SEIRS+V (Susceptible, Exposed, Infectious with multiple sub compartments, Recovered and/or Vaccinated) system with births, deaths, and age structure (see S1 Appendix). For all compartments other than Recovered and/or Vaccinated, event-time distributions were derived from global observations. For Recovered and/or Vaccinated compartments, it was assumed that there would be no waning of infection- or vaccine-derived protection, but birth-death demographic turnover was taken into account.

In the model vaccination operates through preventing infection (and thus disease, but with no impact on breakthrough disease), or on disease (with no impact on infection, but with reduced onward transmission due to shifting symptomatic to asymptomatic cases). In the analysis, the benefits of vaccination are bounded by considering all benefit from protection due to either: prevention of infection (same direct benefit, maximum indirect benefit) versus prevention of disease (same direct benefit, but with minimum indirect benefit). For each scenario, we present ICERs in terms of both infection-preventing and disease-preventing mechanisms.

Contact patterns were estimated from Google mobility data (2021) and the impact of non-pharmaceutical interventions (e.g., lockdowns) using the Oxford Coronavirus Government response tracker [15, 16]. A summary of the epidemiological (including vaccine efficacy) parameters used in the analyses is available in S2 Appendix.

## Model fitting and projections

EpiNow2 [17] was used with early case incidence (from 2020-03-15 to 2020-05-01) to estimate pre-pandemic $R_t$ for the urban population (51.16%, 107,106,007 individuals in 2019) [9], which was then used to set the distribution of the transmission parameter in CovidM by calculating the associated $R_0$. In combination with $R_0$, early deaths (all observed within the first 5 days of the first reported death), were used to determine the number of introductions by assuming an age specific infection fatality rate based on the ENE-COVID Spanish serosurvey study [18], with odds-ratio adjustment of 2.3 as indicated for LMIC settings [19]. The pandemic was projected in the urban population only, using this underlying multiplier as well as changes in Google mobility indices [2]. To match the first peak in observed case incidence, a behaviour-modification parameter (i.e., how much do people further lower their susceptibility and onward transmission in response to the observed cases) was also fitted. Finally, assuming the same baseline transmission parameter distribution and associated behaviour change, the relative transmission of a new variant was fit by assuming an introduction period (2020-12-16 to 2020-12-21), estimating $R_t$ with EpiNow2, and calculating the required multiplier given historical attack rates leading to susceptible depletion.

For vaccine scenarios, it was assumed that the vaccine was infection-blocking and that protection is complete for some individuals and absent in others (i.e., 'all-or-nothing' protection) [20]. Disease-only blocking vaccination scenarios were also analysed. Vaccine doses are distributed among individuals in the Susceptible and Recovered compartments; Susceptible individuals become Vaccinated, and Recovered individuals become Recovered and Vaccinated.

## Health outcomes

The analysis modelled the impact of COVID-19 vaccination on cases, deaths and DALYs. For each scenario, the analysis modelled the health burden in DALYs for symptomatic cases, non-

fatal hospitalisations, non-fatal admissions to critical care, and premature death due to COVID-19.

To estimate the age-specific DALYs for a Covid-19 death in each 5-year age-band of the epidemiological model we applied the modified life-table approach described by Briggs et al. [21] to the United Nations national life tables for Nigeria for the period 2015–2020 [22]. Using this method, the expected discounted future years of quality-adjusted life at each age were calculated using data on the average quality-of-life (QoL) in the general population. Since no data on QoL population norms for Nigeria was available, we instead used data from Zimbabwe [23]. In our base case we assumed that the average baseline mortality and quality amongst individuals who died of Covid-19 was the same as the general population, and future years of life were discounted at an annual rate of 3%.

## Costs

All consequential health sector costs were modelled. We included both the costs of vaccination and the incurred costs of care and treatment of COVID-19. All costs were estimated in 2020 US$. The costing was carried out from a health system perspective using a normative, bottom-up ingredients-based approach.

Our vaccination costs include 12 sub-activities necessary for the planning, roll-out and delivery of vaccines, as well as the cost of the dose itself. These are: planning and coordination, technical assistance, training, social mobilisation, vaccine transport, cold chain, personal protective equipment, hand hygiene, vaccine delivery, vaccination certificates, waste management and pharmacovigilance. The decision to include these 12 sub-activities was based on a model of the costs of delivering COVID-19 vaccine in the 92 COVAX countries developed by UNICEF [24]. Costs were calculated and separated into five input categories: staff salaries, staff per diems, supplies, equipment and vehicles and buildings. Adjustments were made in the costs to account for different cold chain and storage needs per vaccine type. Resource use assumptions, including staff cadre and staff time, were adjusted for each of the three delivery modes modelled. Costs of social mobilization to reach specific age groups or to achieve specific coverage were not included. All scenarios were assumed to be delivered over a 12-month time horizon.

Resource use was estimated through an iterative process. The UNICEF report was reviewed and country-specific relevant sources of unit costs cited for different sub-activities were maintained for analysis. Resource use was obtained from the literature, encompassing both peer-reviewed and grey literature (e.g., Gavi funding applications). These sources were reviewed and a Nigeria-specific qualitative description of plausible resource use was prepared. This description was shared with public health experts in-country (and co-authors on this study) who, through several rounds of validation exercises, arrived at a final resource use description for each sub-activity, including quantities and frequencies of activities both at the national and sub-national level for each of the delivery modalities. See S3 Appendix for details of the resource use assumptions applied in the analysis for each of the 12 sub-activities and the vaccine doses costed. A similar process was followed to determine prices of inputs. An initial list of prices was compiled using the literature and updated to US$ 2020. This list was then reviewed and validated by in-country experts (co-authors CO and BU).

The base price of the dose of the vaccines were set at Nigeria-specific purchasing prices per vaccine type as follows: US$3 for an AstraZeneca-like dose, US$10 for a Johnson & Johnson-like dose, US$19.50 for a Pfizer-BioNTech-like dose, and US$32 for a Moderna-like dose. The base price of the vaccine dose itself was augmented by additional costs due to freight charges, vaccine wastage, and the maintenance of a buffer stock. We assumed an additional 10% freight charge for receiving the vaccine doses in-country from external senders. Domestic

transportation costs are accounted for under the category of 'vaccine transport'. We consulted with vaccine costing experts to discuss uncertainties surrounding an appropriate wastage factor and proportion of buffer stock. We have assumed 15% wastage and the need for a 10% buffer stock, with the cost of these additional doses annuitized over 10 years. Following the iDSI Reference Case [10], a 3% discount rate was used for future costs and for annualising capital investments.

The costing model differentiated between fixed and variable costs by allocating each input to one of three cost centre types: cost per dose, cost per facility per day or cost per country per year. This allowed us to account for economies of scale and observe changes in unit cost depending on the speed and volume of vaccine delivery. Resource use and price data were combined in a Microsoft Excel workbook and unit costs calculated. For policy questions 1 (vaccine type) and 3 (age targeting and coverage), vaccine costs were weighted by likely delivery platform, assuming 50% of vaccines will be delivered through health facilities, 40% through campaigns and 10% through targeted campaigns. Further details on the costs per delivery site per scenario can be requested from the authors.

The economic impact of COVID-19 on the health system includes clinical management. Nigeria-specific costs of COVID-19 care and treatment were used in the analysis [25]. Torres-Rueda et al. (2021) modelled clinical resource use in three base low- and middle-income countries through published sources and expert opinion adjusting for likely resource constraints common in LMICs (e.g. staff shortages in intensive care units). These costs were then extrapolated to other LMICs in the same country-income category using country-specific prices.

## Cost-effectiveness

Cost and health outcome data between intervention and comparator were used to estimate ICERs. To understand possible trade-offs with existing health spending, these estimates of cost-effectiveness were compared against 'supply-side' thresholds which take into account health sector productivity estimated for Nigeria [26], updated to 2020 values (low: 2020 US $364 and high: 2020 US$495). In addition, a threshold of 1 x GDP per capita was also used (2020 US$ 2097), although this is still regarded as aspirational, and not reflective of the actual budget constraint. This threshold can therefore best be regarded as an upper limit in the present analysis.

## Results

We report our findings by policy question. Figures show a disease only impact ('disease') and one that also affects transmission ('infection´). All vaccines are less cost-effective when assuming a disease mechanism vis-à-vis an infection mechanism. The applied cost-effectiveness thresholds are represented as dashed lines in each figure.

### 1. What vaccine to purchase?

We explored the cost-effectiveness of different vaccine types under two scenarios. Figs 1 and 2 present the cost-effectiveness findings assuming 100% coverage for the cohort of 50+ year olds. Figs 3 and 4 show the cost-effectiveness assuming 90% coverage for all adults 18+ years old. Viral vector vaccines are cost-effective (or even cost saving) by all thresholds and both vaccine mechanisms assuming vaccination only of all older adults. They are cost-effective in relation to all thresholds when vaccinating 90% of all adults, but only when assuming an infection mechanism. Vaccines using mRNA technologies are not cost-effective under any threshold or vaccine mechanism when assuming vaccination of 90% of adults. When vaccinating all older

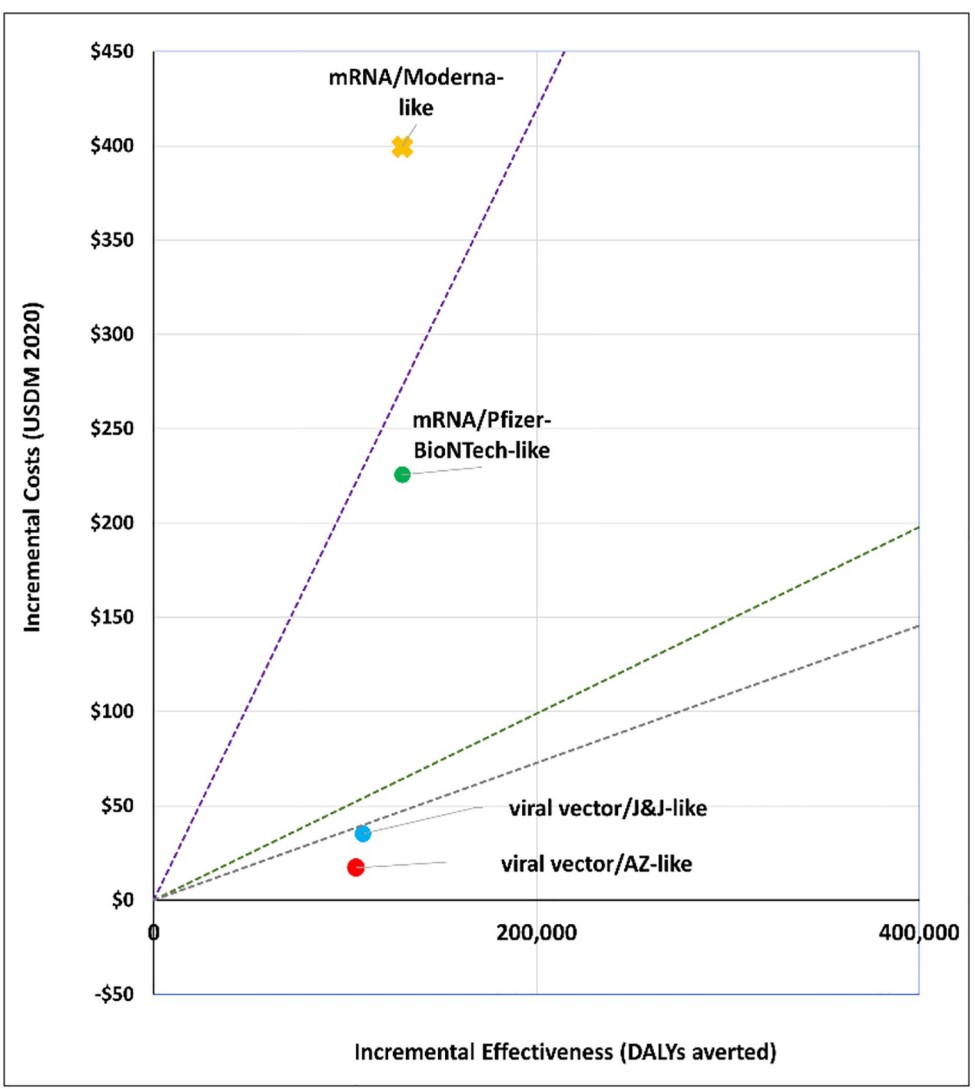

**Fig 1. Scenario 1: Cost-effectiveness analysis of vaccine type assuming 100% coverage for adults aged 50+ years old: Disease vaccine mechanism.** Applied cost-effectiveness thresholds represented by the dashed lines.

adults, a Pfizer-like vaccine would be cost effective assuming both vaccine mechanisms but only through applying a 1x GDP per capita threshold.

## 2. How should vaccines be delivered?

We estimated the cost-effectiveness of three methods of vaccine delivery through health facilities (HF), campaigns (C) and targeted campaigns (TC) for a viral vector vaccine (AstraZeneca-like) and an mRNA vaccine (Moderna-like). We focused on two scenarios: 100% coverage of all adults 50+ (Figs 5 and 6) and 25% coverage of all adults but prioritising all 50+ year olds (Figs 7 and 8). We found that the different vaccine delivery methods have little effect on cost-effectiveness. For both scenarios, the mRNA vaccine, independent of delivery method, is only cost-effective under a 1x GDP per capita threshold assuming an infection vaccine mechanism. A viral vector vaccine, delivered through any method, is cost-effective in both scenarios by all

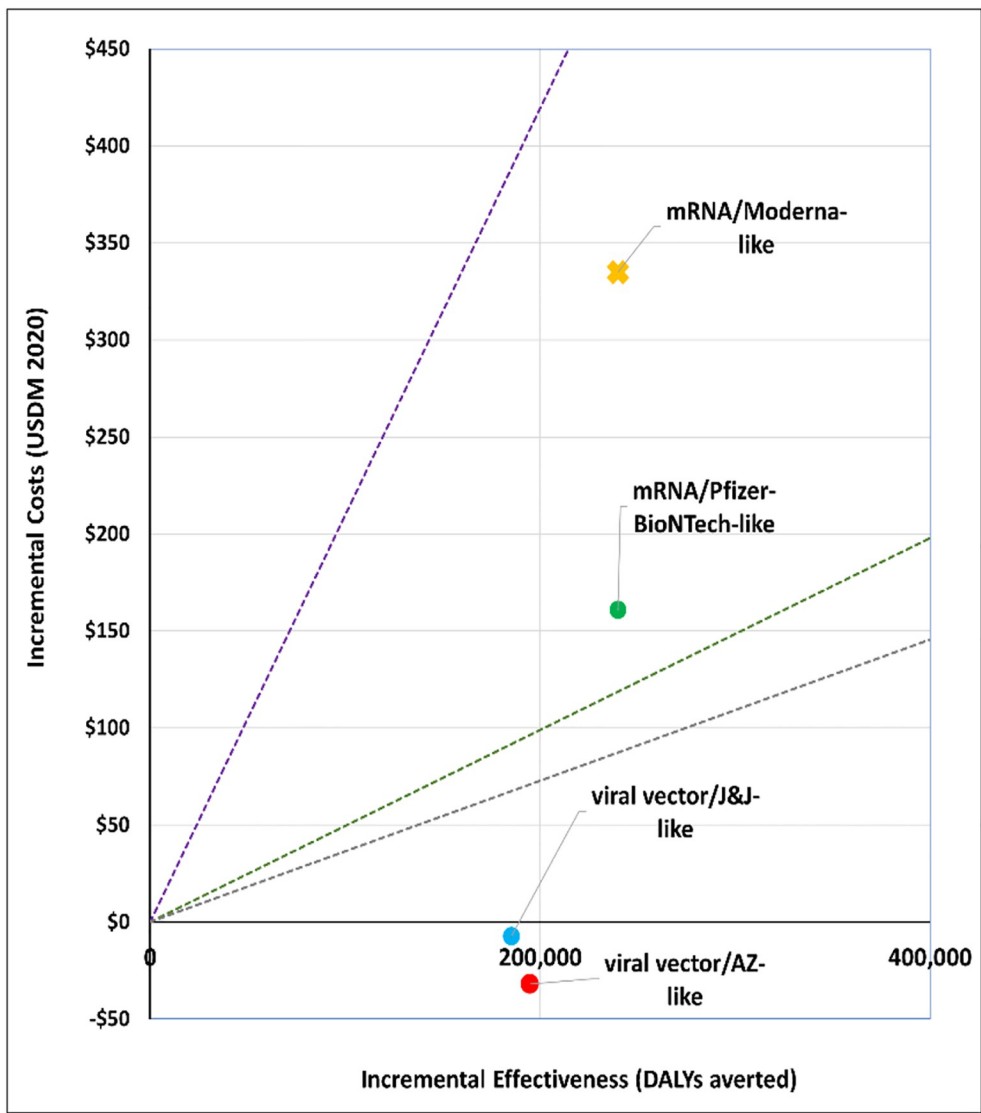

**Fig 2. Scenario 1: Cost-effectiveness analysis of vaccine type assuming 100% coverage for adults aged 50+ years old: Infection vaccine mechanism.** Applied cost-effectiveness thresholds represented by the dashed lines.

thresholds when assuming a disease vaccine mechanism and cost-saving when assuming an infection mechanism.

## 3. Who should be prioritized?

We modelled five age targeting and coverage scenarios for a viral vector vaccine (AstraZeneca-like) and an mRNA vaccine (Moderna-like). See Figs 9 to 12. For a viral vector vaccine, targeting 70% or 100% of 50+ year olds, as well as targeting 25% of all adults prioritising 50+ year olds, is either cost-effective (by all thresholds) or cost-saving irrespective of the vaccine mechanism applied. Vaccinating either 90% of all adults, or 70% of 50+ year olds and 25% of 18–49 year olds, are comparatively less cost-effective. For an mRNA vaccine, no strategy is cost-effective when applying the Ochalek supply-side based thresholds. Assuming an infection

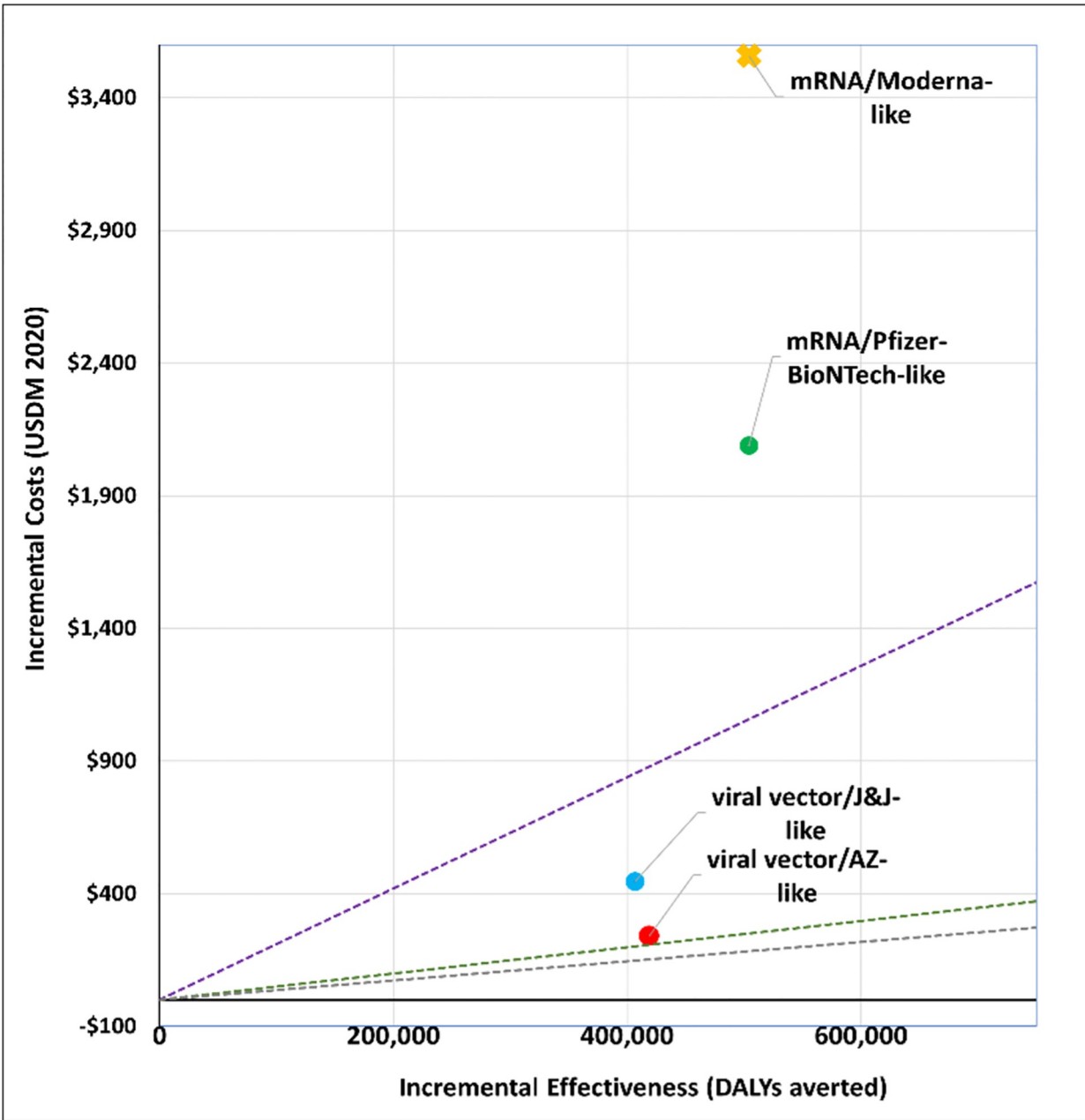

**Fig 3. Scenario 2: Cost-effectiveness analysis of vaccine type assuming 90% coverage for all adults aged 18+ years old: Disease vaccine mechanism.** Applied cost-effectiveness thresholds represented by the dashed lines.

mechanism, all age and coverage strategies are cost-effective by the 1x GDP per capita threshold except for vaccinating 90% of all adults (Fig 12).

Costs, DALYs averted and ICERs for each scenario are presented in S4 Appendix.

## Discussion

COVID-19 vaccines can be highly effective and cost-effective (as well as cost-saving), although an important determinant of the latter is the price per dose and the age groups prioritised for

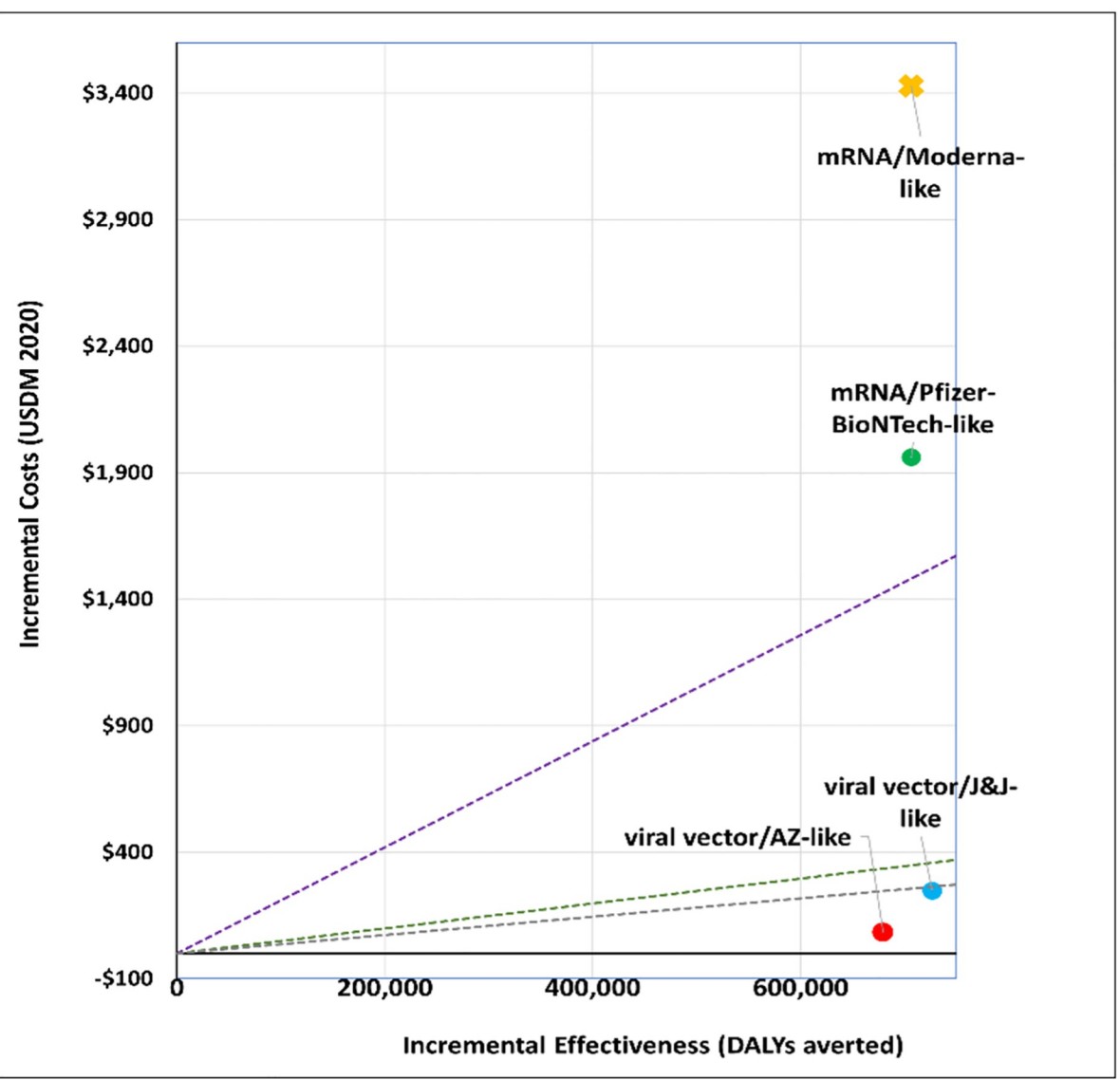

**Fig 4. Scenario 2: Cost-effectiveness analysis of vaccine type assuming 90% coverage for all adults aged 18+ years old: Infection vaccine mechanism.** Applied cost-effectiveness thresholds represented by the dashed lines.

vaccination. Our analysis suggests that viral vector vaccines (similar to those produced by AstraZeneca and Johnson & Johnson) may represent more cost-effective choices from the Nigerian health system perspective. Higher prices may be justifiable from a societal perspective, but if funds are being drawn from current health budgets, vaccines priced under US$ 10 per dose compare favourably with other technologies that could be provided within the health budget.

It has been reported recently that Moderna has reached an agreement with the African Union to supply its COVID-19 vaccines at a price of US$ 7 dollars per dose [27]. This would bring it more in line with the costs of the AstraZeneca vaccine, and most likely substantially improve its relative cost-effectiveness. Notably, similar analyses undertaken to explore the cost-effectiveness of vaccination against COVID-19 in Sindh province in Pakistan [2] highlighted the impact of vaccine price on cost-effectiveness, suggesting the importance of not

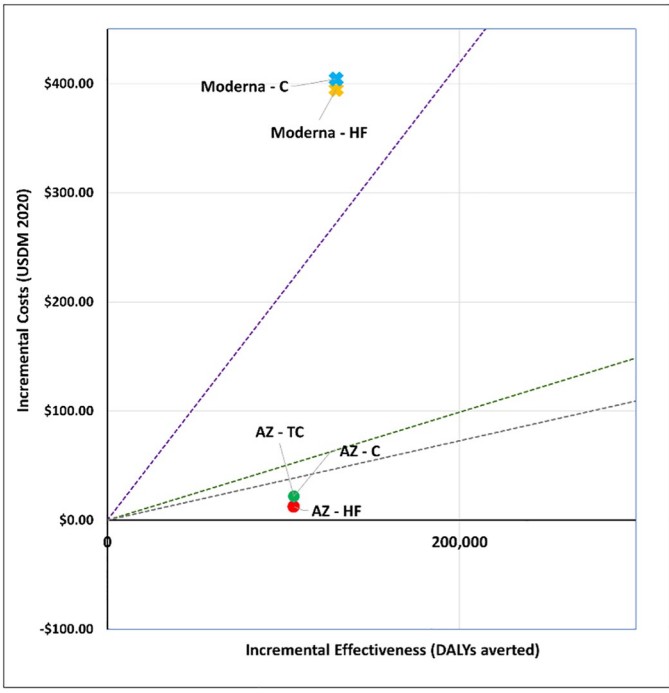

**Fig 5. Scenario 3: Cost-effectiveness analysis of vaccine delivery types assuming 100% coverage of all adults aged 50+ years old: Disease vaccine mechanism.** Applied cost-effectiveness thresholds represented by the dashed lines.

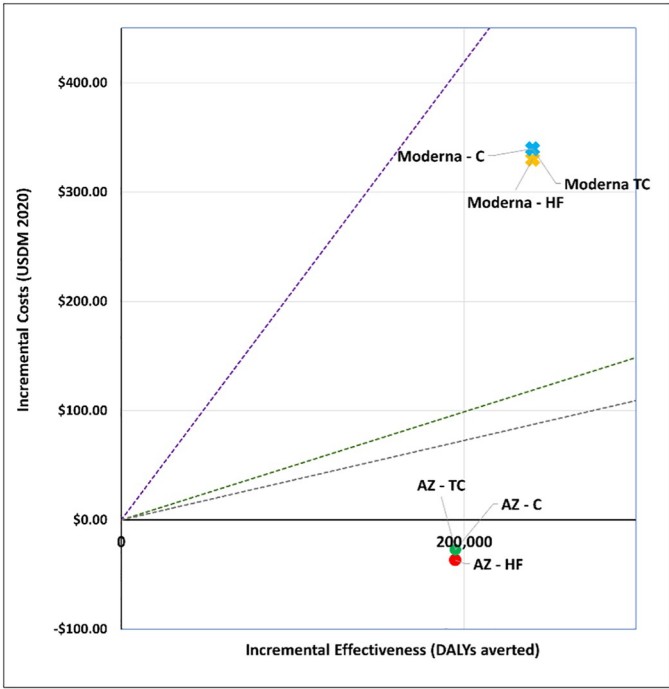

**Fig 6. Scenario 3: Cost-effectiveness analysis of vaccine delivery types assuming 100% coverage of all adults aged 50+ years old: Infection vaccine mechanism.** Applied cost-effectiveness thresholds represented by the dashed lines.

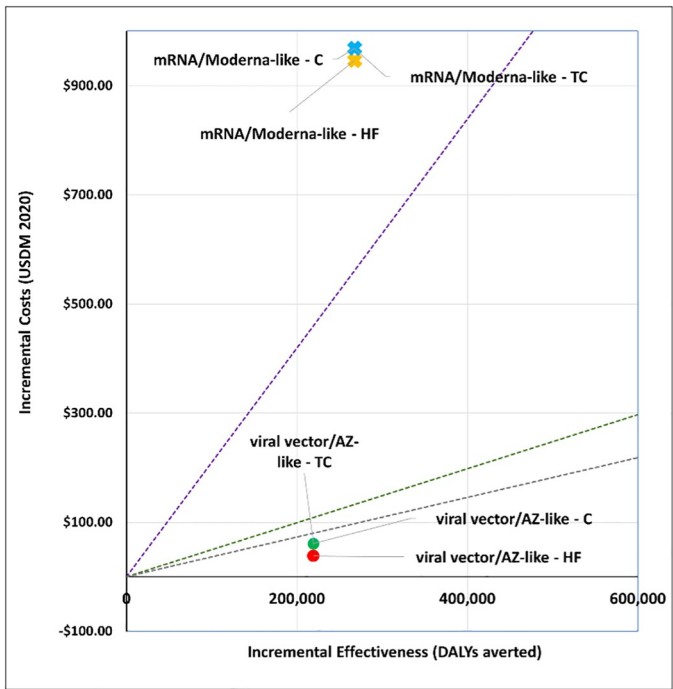

**Fig 7. Scenario 4: Cost-effectiveness analysis of vaccine delivery types assuming 25% coverage of all adults first prioritising all 50+ year olds: Disease vaccine mechanism.** Applied cost-effectiveness thresholds represented by the dashed lines.

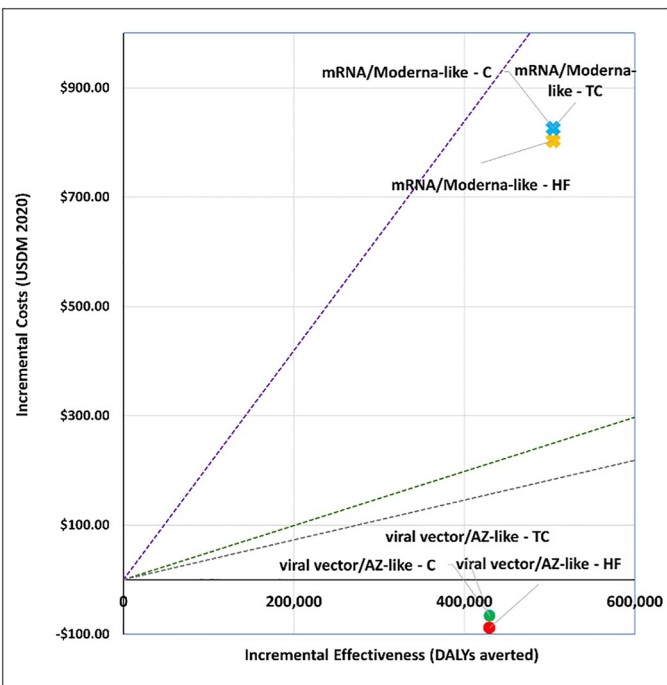

**Fig 8. Scenario 4: Cost-effectiveness analysis of vaccine delivery types assuming 25% coverage of all adults first prioritising all 50+ year olds: Infection vaccine mechanism.** Applied cost-effectiveness thresholds represented by the dashed lines.

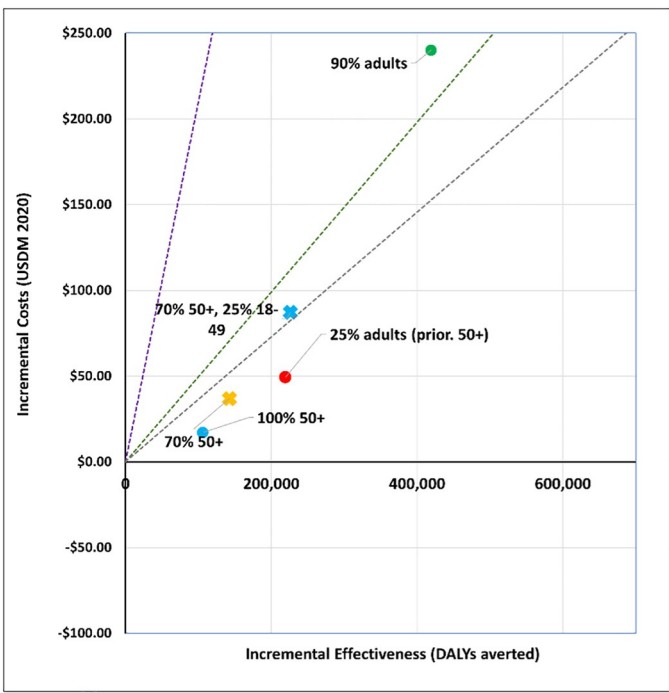

**Fig 9. Scenario 5: Cost-effectiveness analysis of age prioritization and target coverage for a viral vector vaccine (AstraZeneca-like): Disease vaccine mechanism.** Applied cost-effectiveness thresholds represented by the dashed lines.

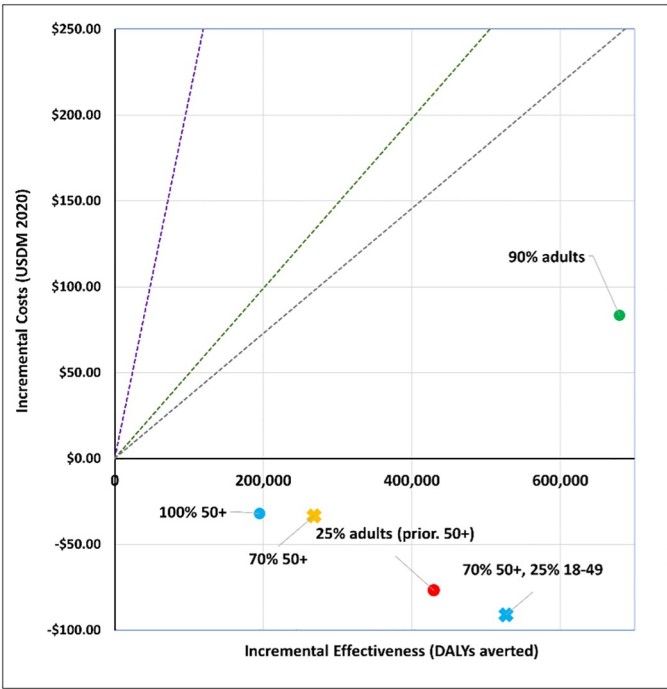

**Fig 10. Scenario 5: Cost-effectiveness analysis of age prioritization and target coverage for a viral vector vaccine (AstraZeneca-like): Infection vaccine mechanism.** Applied cost-effectiveness thresholds represented by the dashed lines.

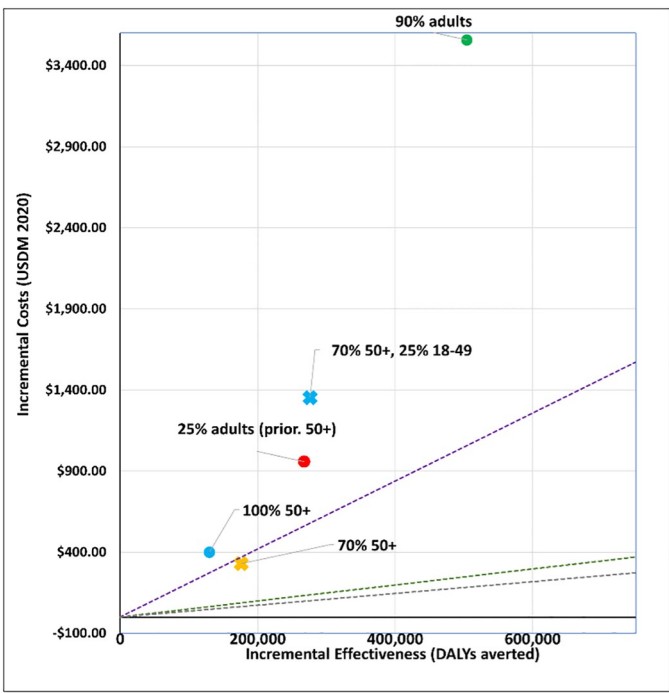

**Fig 11. Scenario 6: Cost-effectiveness analysis of age prioritization and target coverage for an mRNA vaccine (Moderna-like): Disease vaccine mechanism.** Applied cost-effectiveness thresholds represented by the dashed lines.

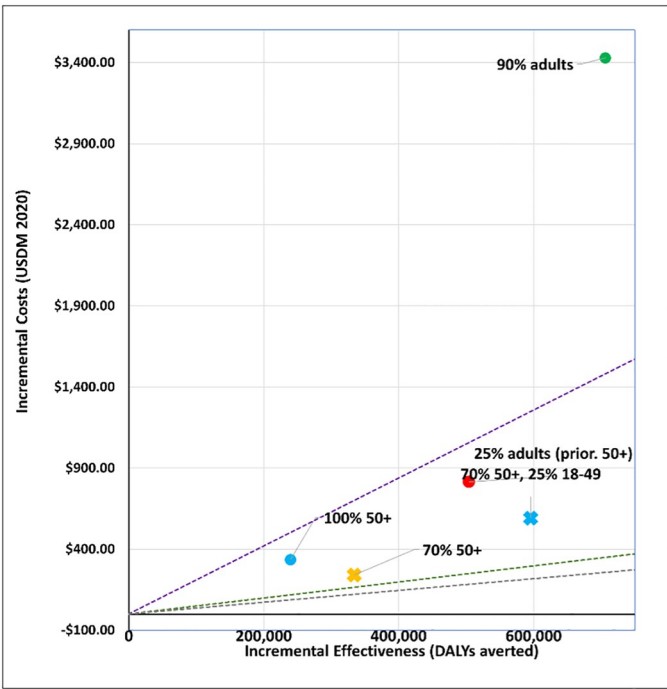

**Fig 12. Scenario 6: Cost-effectiveness analysis of age prioritization and target coverage for an mRNA vaccine (Moderna-like): Infection vaccine mechanism.** Applied cost-effectiveness thresholds represented by the dashed lines.

exceeding US$ 10 per dose, and preferably keeping it $US 6 or less, to ensure that it is cost-effective from a health system perspective.

The cost-effectiveness analysis broadly confirms the age group prioritisation strategy of the Nigerian government (which focused on a 50+ cohort during phase 2 of the roll-out) and that different types of delivery make little difference to the results. Notably recent evidence suggests that if countries wish to minimise deaths, prioritising the vaccination of senior adults (65+) was the optimal strategy [28]. A cost-effectiveness analysis of COVID-19 vaccination in Kenya found that vaccinating younger adults (under 50 years of age was unlikely to be cost-effective [29]) when applying a locally relevant cost-effectiveness threshold, although speed of roll out was also important in terms of averting deaths and overall costs. In that analysis, it was assumed that vaccines were procured as US$7 per dose, and delivery costs ranged from US$ 3.90 to US$ 6.11 per dose.

The modelling also provided estimates of the total economic costs of the vaccination strategies broken down by year for a 5-year period (See S5 Appendix). These total costs included the costs of vaccinations themselves plus care and treatment of patients. These costs are also estimated for the two different vaccine mechanism assumptions. Unlike financial costs, these costs aim to take into account the full (opportunity) cost of providing a service and include estimates of staff time even though, for example, employee salaries would have been paid in any case. Most vaccination-related costs are concentrated in year 1, with costs at much lower levels between years 2 and 5. Similar findings were seen for all the vaccines in our analysis in terms of how the costs were distributed over the 5-year period, although overall magnitudes differed. For the AZ-like vaccine, year 1 costs were estimated to be around $480 million (vaccinating 25% of the population and assuming a disease-only vaccine mechanism) In contrast, for the Moderna-like vaccine, costs were estimated to reach as high as $1,406 million under the same scenario assumptions.

While we have taken into account costs of social mobilization, there is considerable uncertainty about the effectiveness and costs of ancillary interventions to address vaccine hesitancy, which is a problem in many regions of the world. Evidence from the African continent suggests that hesitancy has been linked to concerns about vaccine safety, side effects, and effectiveness, which have been found to be widespread even among health workers [30–33]. Addressing vaccine hesitancy is likely to be important in achieving some of the high coverages that we have modelled. The costs of such interventions were however not analysed in the present study.

The modelling presented above focused only on a health system perspective. It is likely that a broader societal perspective would have highlighted the wider benefits of vaccination. Pearson et al. (2021) found that COVID-19 vaccination in Sindh province, Pakistan was cost-saving from a societal perspective [2]. The aim with the Nigerian analysis is to help country policymakers navigate a path of managing the COVID-19 epidemic while accounting for the longer-term sustainability of the health sector, which largely bears the majority of the costs in preventing and treating COVID-19 cases, even when vaccines are donated. The present analysis assumed a 12-month vaccination roll-out and no vaccine waning in the base-case: if as is likely, additional doses are needed for those already vaccinated on an ongoing (multi-year) basis, accounting for health system constraints and the risks to other essential health services becomes even more important. For those reasons it is arguably important to take a more 'conservative' view and focus only on a health system perspective.

Nevertheless, it is clear that the economic benefits to vaccinating against COVID-19 reach beyond the healthcare sector. The first year of the pandemic saw a huge shock to the world economy, with economic contractions that were three times worse than during the 2008–09 financial crisis [34]. This pandemic induced financial shock can be reduced by vaccination.

Some estimates for the global value to the economy by increasing vaccine supply are as large as 5,800 USD per course, or 576 to 989 USD for speeding up vaccination by four months [35]. This greatly dwarfs the global price. A large part of the economic benefits arising from these estimates are based on the removal of non-pharmaceutical interventions such as lockdowns; these have for the most part been removed in Nigeria.

While there is insufficient evidence on the macro-economic impact of vaccination in LMIC settings to know what the benefits would be for countries like Nigeria, it is also apparent that there are wider global health security concerns that should weigh on decisions made by high income countries with respect to supporting vaccination in poorer countries. African countries are working to procure COVID-19 vaccines directly, but for over half of them the estimated cost of vaccination would exceed the total government health expenditure, potentially jeopardizing the financing of essential health services [8].

Our study is subject to a number of limitations. There is a lack of evidence on immunisation waning, either from natural infection or vaccination. Waning was not accounted for in the model and therefore cost-effectiveness may be over-estimated. The epidemiological model does not take into account the rapid emergence of new variants. The model does not account for future changes in the technologies available in the prevention or management of disease, or other changes that may affect the epidemiology and the impact on outcomes. Our analyses take a health system perspective only and therefore cost-effectiveness may be under-estimated. The study does not take into account health system constraints in the delivery of the vaccine (such as human resource availability), nor does it include the displacement of health services as a consequence of resources being redirected to mass vaccination. This may result in cost-effectiveness being over-estimated. Lastly, we did not adjust costs of delivery (e.g. social mobilisation costs) as coverage increases and more intense activities need to be undertaken to address hesitancy and vaccinate the remaining population.

The evidence presented in this report including the accompanying cost-effectiveness analyses are subject to uncertainty which could be further addressed by further research and analytical work. Some options that could be taken forward include: understanding the costs of addressing vaccine hesitancy, broadening the analysis to include a societal perspective, including new variants and booster doses, and accounting for health system constraints.

The findings from our study could be extrapolated to other settings; countries purchasing vaccinations against COVID-19 should ensure value for money when choosing among alternatives. As is the case in similar studies based in LMICs, we find that price per dose will probably be an important determinant of cost-effectiveness, as well as the group prioritised for vaccination [2, 29]. How vaccines are delivered to eligible populations should be based on the most feasible arrangements available in the country as the cost-effectiveness of different delivery methods is similar. Finally, this work shows the viability, and indeed desirability, of developing combined epidemiological and economic models in these settings, although further work is needed in aligning methods when, for example, considering equity [36].

## Conclusions

COVID-19 vaccines can be highly effective and cost-effective (as well as cost-saving), although an important determinant of the latter is the price per dose and the age groups prioritised for vaccination. From a health system perspective, viral vector vaccines (similar to those produced by AstraZeneca and Johnson & Johnson) may represent more cost-effective choices for Nigeria at current prices, although this may change with price negotiation. The method of delivery of vaccines has little effect on the cost-effectiveness of the vaccine options. Age targeting, to older

adults, will likely be more cost-effective. Uncertainty remains on the additional costs needed to address vaccine hesitancy.

## Supporting information

**S1 Appendix. Epidemiological model outline.**
(DOCX)

**S2 Appendix. Summary of the epidemiological parameters (including vaccine efficacy) used in the analyses.**
(DOCX)

**S3 Appendix. Resource use assumptions used in the analysis.**
(DOCX)

**S4 Appendix. Costs, DALYs averted and ICERs per modelled scenario.**
(DOCX)

**S5 Appendix. Total vaccination and COVID-19 management costs over 5 years.**
(DOCX)

## Acknowledgments

The authors acknowledge the contribution of Dr Nicholas G. Davies and Dr Rosanna C. Barnard for their contribution to the development of the general CovidM framework. We also sincerely acknowledge the Nigerian Federal Ministry of Health and the National Primary Health Care Development Agency for their inputs and support of this work; Dr Justice Nonvignon and Dr Elias Esfaw at Africa CDC for their inputs into this project; Dr Tom Drake at CGD for his support and overseeing iDSI's series of COVID-19 vaccine HTA projects.

## Author Contributions

**Conceptualization:** Francis J. Ruiz, Carl A. B. Pearson, Mark Jit, Anna Vassall, Benjamin S. C. Uzochukwu.

**Data curation:** Sergio Torres-Rueda, Carl A. B. Pearson.

**Formal analysis:** Sergio Torres-Rueda, Carl A. B. Pearson, Eleanor Bergren, Simon R. Procter, Andres Madriz-Montero.

**Funding acquisition:** Mark Jit, Anna Vassall.

**Investigation:** Sergio Torres-Rueda, Carl A. B. Pearson, Eleanor Bergren, Chinyere Okeke, Simon R. Procter, Andres Madriz-Montero, Benjamin S. C. Uzochukwu.

**Methodology:** Sergio Torres-Rueda, Carl A. B. Pearson, Eleanor Bergren, Mark Jit, Anna Vassall.

**Project administration:** Francis J. Ruiz, Sergio Torres-Rueda, Mark Jit, Anna Vassall.

**Software:** Carl A. B. Pearson.

**Supervision:** Francis J. Ruiz, Mark Jit, Anna Vassall, Benjamin S. C. Uzochukwu.

**Visualization:** Francis J. Ruiz, Carl A. B. Pearson, Chinyere Okeke, Andres Madriz-Montero.

**Writing – original draft:** Francis J. Ruiz, Sergio Torres-Rueda.

**Writing – review & editing:** Carl A. B. Pearson, Eleanor Bergren, Chinyere Okeke, Simon R. Procter, Andres Madriz-Montero, Mark Jit, Anna Vassall, Benjamin S. C. Uzochukwu.

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
