## [Decision Letter · Decision Letter 0]

20 Oct 2022

PGPH-D-22-01313

What, how and who: cost-effectiveness analyses of COVID-19 vaccination to inform key policies in Nigeria

Dear Dr. Francis Ruiz,

Thank you for submitting your manuscript to PLOS Global Public Health. After careful consideration, we feel that it has merit but does not fully meet PLOS Global Public Health’s publication criteria as it currently stands. Therefore, we invite you to submit a revised version of the manuscript that addresses the points raised during the review process.

We look forward to receiving your revised manuscript.

Kind regards,

Genevieve Cecilia Aryeetey, Ph.D

Academic Editor

Journal Requirements:

1. We have amended your Competing Interest statement to comply with journal style. We kindly ask that you double check the statement and let us know if anything is incorrect. 

2.Please amend your detailed Financial Disclosure statement. This is published with the article. It must therefore be completed in full sentences and contain the exact wording you wish to be published.

a. State what role the funders took in the study. If the funders had no role in your study, please state: “The funders had no role in study design, data collection and analysis, decision to publish, or preparation of the manuscript.”

b. If any authors received a salary from any of your funders, please state which authors and which funders.

3. In the online submission form, you indicated that your data will be submitted to a repository upon acceptance.  We strongly recommend all authors deposit their data before acceptance, as the process can be lengthy and hold up publication timelines. Please note that, though access restrictions are acceptable now, your entire data will need to be made freely accessible if your manuscript is accepted for publication. This policy applies to all data except where public deposition would breach compliance with the protocol approved by your research ethics board. If you are unable to adhere to our open data policy, please kindly revise your statement to explain your reasoning and we will seek the editor's input on an exemption. Please be assured that, once you have provided your new statement, the assessment of your exemption will not hold up the peer review process.

Additional Editor Comments (if provided):

Reviewers' comments:

Reviewer's Responses to Questions

**Comments to the Author**

1. Does this manuscript meet PLOS Global Public Health’s publication criteria? Is the manuscript technically sound, and do the data support the conclusions? The manuscript must describe methodologically and ethically rigorous research with conclusions that are appropriately drawn based on the data presented.

Reviewer #1: Yes

Reviewer #2: Yes

2. Has the statistical analysis been performed appropriately and rigorously?

Reviewer #1: Yes

Reviewer #2: Yes

3. Have the authors made all data underlying the findings in their manuscript fully available (please refer to the Data Availability Statement at the start of the manuscript PDF file)?

Reviewer #1: Yes

Reviewer #2: Yes

4. Is the manuscript presented in an intelligible fashion and written in standard English?

Reviewer #1: Yes

Reviewer #2: Yes

5. Review Comments to the Author

Reviewer #1: Ethics statement: An ethics statement is a one-page essay that gives a picture of your core values and what potential supervisors, colleagues, or clients can expect from working with you. These must be clearly stated or mentioned

Ethical considerations in research are a set of principles that guide your research designs and practices. These principles include voluntary participation, informed consent, anonymity, confidentiality, potential for harm, and results communication.--- So provide the ethical considerations in this research

Objective: The objective of this study was not clearly stated

Discussion

Second statement: Taking a health system perspective only would suggest that viral vector vaccines (similar to those produced by AstraZeneca and Johnson & Johnson) may represent more cost-effective choices from the Nigerian perspective.

‘’ONLY’’ SHOULD BE CHANGE TO ‘’ONE’’

Similar findings are seen for all the vaccines------ This statement needs to be referenced

Conclusion

The mode of delivery of vaccines has little effect on the cost effectiveness of the vaccine options------- Provide the analysis/table that proves or supports this statement.

References

Vancouver reference style was used in the text while APA style was used in the list of reference. This is a contradiction. In Vancouver reference style, et al comes after listing 6 authors and their initials. All references in the list of reference must be re-written to conform with Vancouver reference style

Implications of the study: This should be properly stated

Contribution to body of knowledge: The study is so germane for contribution to knowledge to be omitted

Reviewer #2: Language; simple and clear, no grammatical errors

Flow; coherent and logical flow of information in all sections.

Literature review; extensive and relevant

Methods, Data analysis; detailed, focused, contains all the necessary information. The choice of models was relevant to the topic

Results and discussion; all important elements as guided by analysis frameworks were presented, and discussed.

Conclusions; in line with the main results

Usefulness of the study; very relevant to the contexts of LMIC

6. PLOS authors have the option to publish the peer review history of their article (what does this mean?). If published, this will include your full peer review and any attached files.

**Do you want your identity to be public for this peer review?** For information about this choice, including consent withdrawal, please see our Privacy Policy.

Reviewer #1: **Yes: **Ibraheem Shola Abdulraheem

Reviewer #2: No

---

## [Decision Letter · Decision Letter 1]

15 Feb 2023

What, how and who: cost-effectiveness analyses of COVID-19 vaccination to inform key policies in Nigeria

PGPH-D-22-01313R1

Dear Mr Ruiz,

We are pleased to inform you that your manuscript 'What, how and who: cost-effectiveness analyses of COVID-19 vaccination to inform key policies in Nigeria' has been provisionally accepted for publication in PLOS Global Public Health.

Best regards,

Julia Robinson

Executive Editor

Reviewer Comments (if any, and for reference):

Reviewer's Responses to Questions

**Comments to the Author**

1. If the authors have adequately addressed your comments raised in a previous round of review and you feel that this manuscript is now acceptable for publication, you may indicate that here to bypass the “Comments to the Author” section, enter your conflict of interest statement in the “Confidential to Editor” section, and submit your "Accept" recommendation.

Reviewer #2: All comments have been addressed

2. Does this manuscript meet PLOS Global Public Health’s publication criteria? Is the manuscript technically sound, and do the data support the conclusions? The manuscript must describe methodologically and ethically rigorous research with conclusions that are appropriately drawn based on the data presented.

Reviewer #2: Yes

3. Has the statistical analysis been performed appropriately and rigorously?

Reviewer #2: Yes

4. Have the authors made all data underlying the findings in their manuscript fully available (please refer to the Data Availability Statement at the start of the manuscript PDF file)?

Reviewer #2: Yes

5. Is the manuscript presented in an intelligible fashion and written in standard English?

Reviewer #2: Yes

6. Review Comments to the Author

Reviewer #2: Authors have adequately responded to all comments given by primary reviewers

7. PLOS authors have the option to publish the peer review history of their article (what does this mean?). If published, this will include your full peer review and any attached files.

**Do you want your identity to be public for this peer review?** For information about this choice, including consent withdrawal, please see our Privacy Policy.

Reviewer #2: No
